# Beyond Nitrogen in the Oxygen Reduction Reaction on Nitrogen-Doped Carbons: A NEXAFS Investigation

**DOI:** 10.3390/nano11051198

**Published:** 2021-05-01

**Authors:** Eugenia Tanasa, Florentina Iuliana Maxim, Tugce Erniyazov, Matei-Tom Iacob, Tomáš Skála, Liviu Cristian Tanase, Cătălin Ianăși, Cristina Moisescu, Cristina Miron, Ioan Ardelean, Vlad-Andrei Antohe, Eugenia Fagadar-Cosma, Serban N. Stamatin

**Affiliations:** 1Faculty of Applied Chemistry and Materials Science, Politehnica University of Bucharest, Splaiul Independentei Str. No. 313, 060042 Bucharest, Romania; eugenia.tanasa@physics.pub.ro; 2Nano-SAE Research Centre, University of Bucharest, Atomistilor 405, 077125 Magurele, Ilfov, Romania; iuliana.maxim@3nanosae.org (F.I.M.); tugcegoktas95@gmail.com (T.E.); tom.iacob@3nanosae.org (M.-T.I.); 3Department of Surface and Plasma Science, Charles University, V Holešovičkách 2, 18000 Prague, Czech Republic; tomas.skala@elettra.eu; 4National Institute of Materials Physics, Atomistilor 405A, 077125 Magurele, Ilfov, Romania; liviu.tanase@infim.ro; 5“Coriolan Drăgulescu” Institute of Chemistry, Mihai Viteazul Ave. 24, 300223 Timisoara, Romania; cianasic@yahoo.com (C.I.); efagadarcosma@acad-icht.tm.edu.ro (E.F.-C.); 6Department of Microbiology, Institute of Biology Bucharest, Splaiul Independenței 296, 060031 Bucharest, Romania; cristina.moisescu@yahoo.com (C.M.); ioan.ardelean@ibiol.ro (I.A.); 7Faculty of Physics, University of Bucharest, Atomistilor 405, 077125 Magurele, Ilfov, Romania; cristina.miron@fizica.unibuc.ro (C.M.); vlad.antohe@fizica.unibuc.ro (V.-A.A.); 8Institute of Condensed Matter and Nanosciences (IMCN), Université catholique de Louvain (UCLouvain), Place Croix du Sud 1, B-1348 Louvain-la-Neuve, Belgium

**Keywords:** nitrogen-doped carbon, structure, oxygen reduction reaction

## Abstract

Polymer electrolyte membrane fuel cells require cheap and active electrocatalysts to drive the oxygen reduction reaction. Nitrogen-doped carbons have been extensively studied regarding their oxygen reduction reaction. The work at hand looks beyond the nitrogen chemistry and brings to light the role of oxygen. Nitrogen-doped nanocarbons were obtained by a radio-frequency plasma route at 0, 100, 250, and 350 W. The lateral size of the graphitic domain, determined from Raman spectroscopy, showed that the nitrogen plasma treatment decreased the crystallite size. Synchrotron radiation photoelectron spectroscopy showed a similar nitrogen chemistry, albeit the nitrogen concentration increased with the plasma power. Lateral crystallite size and several nitrogen moieties were plotted against the onset potential determined from oxygen reduction reaction curves. There was no correlation between the electrochemical activity and the sample structure, as determine from Raman and synchrotron radiation photoelectron spectroscopy. Near-edge X-ray absorption fine structure (NEXAFS) was performed to unravel the carbon and nitrogen local structure. A difference analysis of the NEXAFS spectra showed that the oxygen surrounding the pyridinic nitrogen was critical in achieving high onset potentials. The work shows that there were more factors at play, other than carbon organization and nitrogen chemistry.

## 1. Introduction

To achieve sustainability, there is a need for devices that can store or convert renewable energy. Electrochemical devices can store renewable energy in batteries or use the renewable energy in electrolyzers to generate alternative fuels, such as hydrogen, which can be used in fuel cells. To balance the charge in such electrochemical devices, a cathode exposed to air is used for electrochemical reduction reactions. The oxygen reduction reaction (ORR) is the key reaction in air–cathode electrochemical devices: polymer electrolyte membrane fuel cells [1], metal–air batteries [2], and microbial fuel cells [3]. Platinum-group metals and their alloys are the electrocatalysts of choice in the oxygen reduction reaction [1]. Nitrogen-coordinated metals (e.g., Fe, Co) have been shown to drive the oxygen reduction reaction at similar overpotentials to that of platinum [4,5]. To further decrease the cost, heteroatom-doped nanocarbons have shown remarkable progress in terms of onset potential and current density [6].

The ORR active sites on metal nanoparticles are restricted to the metal facets, which makes it less challenging to identify active sites and understand reaction mechanisms on metal surfaces than on heteroatom-doped carbons [7]. The quest for ORR active sites on heteroatom-doped carbons is severely hindered by the challenges in characterizing heteroatom local structure, carbon organization, and the influence of other dopants (e.g., oxygen, inherently present in carbon materials). For example, testing reaction mechanisms by in situ X-ray probing techniques (attainable for metallic electrocatalysts) is challenging to perform for light elements due to the additional vacuum needed to extract photoelectrons. Such experimental challenges in characterizing nanocarbons has impeded the efforts to reach a consensus on the nature of active sites in heteroatom-doped carbons.

Nitrogen has been considered the active electrocatalytic center in nitrogen-doped carbons [8,9]. Nitrogen bonds to carbon in different configurations. Some nitrogen atoms are located at the edge of the carbon network (pyridinic and pyrrolic). Other nitrogen atoms are bonded to three carbons and located within the carbon network (graphitic nitrogen). There is an ongoing debate in the literature on the role of pyridinic, pyrrolic, or graphitic nitrogen in ORR [8,10,11,12,13]. Recent studies showed that graphitic and pyridinic nitrogen can work in synergy to drive ORR [14].

However, there are more factors at play in the ORR on nitrogen-doped carbons. For a given pH value [12], ORR activity on nitrogenated carbon electrodes is governed by: (1) the electron-donating ability of the electrode—influenced by the nitrogen concentration [15]; (2) oxygen adsorption and desorption from the surface, influenced by the nitrogen surface chemistry [9,13,16] (i.e., type of nitrogen); (3) textural properties such as pore size and effective surface area, which are directly proportional to the number of active sites [17], and (4) dimension of the crystallites influenced by the carbon organization [18,19,20,21]. Recent theoretical studies suggested that oxygen does not bind to nitrogen atoms [22], and a similar mechanism is expected for all heteroatom-doped carbons. The role of other heteroatoms (other than nitrogen) in oxygen reduction reaction activity has been studied [23,24]. Nanocarbons are inherently doped with oxygen. Therefore, traditional heteroatom-doped carbons are dual-doped. A secondary atom has been shown to break the well-known scaling relationships [25]. There are no studies to date that simultaneously characterize the inherent heteroatom (that is, oxygen) and the secondary heteroatom (that is, nitrogen, phosphorous, boron, etc.). Such studies are difficult, as standard characterization tools focus on one heteroatom at a time.

Herein, we bring to light a new component that is ORR-active and is inherently present on the surface of carbons; that is, oxygen functionalities. To analyze the ORR with respect to the nitrogen concentration and type, we have conducted a combined photoelectron spectroscopy and near-edge X-ray absorption fine structure (NEXAFS). Textural properties were characterized by nitrogen adsorption isotherms. Raman analysis was used to characterize the carbon structure. The onset potential was plotted against nitrogen concentration, nitrogen surface chemistry, and carbon organization properties. Postmortem Raman analysis was used to investigate the role of carbon organization. Based on the results, we concluded that oxygen and nitrogen moieties were working in synergy to drive ORR.

## 2. Experimental

### 2.1. Reagents

Carbon Vulcan XC-72R (College Station, TX, USA) and KOH (Cristal R Chim, Bucharest, Romania) were used without further purification. Ultrapure water (>18.2 MΩ cm) was supplied by Milli-Q DirectQ System (Burlington, MA, USA).

### 2.2. Nitrogen Radiofrequency Plasma Treatment

The nitrogen doping of carbon samples was carried out using a home-built radio-frequency plasma system with a 2.46 MHz plasma frequency. To surface-treat nanocarbon powders, 100 mg commercial Vulcan XC-72R (herein Vulcan) was mixed with 5 mL acetone (S.C. Chimreactiv S.R.L., Bucharest, Romania) then placed in a quartz crucible. The solution formed a thin film upon drying at 130 °C. Quartz crucibles containing nanocarbons were placed inside the quartz tube reactor of the RF plasma system. The base pressure was kept below 10^−2^ mbar. The tube was then filled with nitrogen at 100 mL/min. The plasma treatment took place at 5 × 10^−1^ mbar. Nitrogen plasma was formed inside the quartz tube by applying 100, 250, and 350 W RF power for 60 min to convert Vulcan to CN100, CN250, and CN350, respectively.

### 2.3. Photoelectron Spectroscopy and X-ray Absorption Spectroscopy

The Materials Science Beamline (bending-magnet beamline; tuning range 22–1000 eV) at Elettra Synchrotron (Trieste, Italy) was used to carry out the photoemission and X-ray absorption experiments. The samples were characterized at base pressure below 2 × 10^−10^ mbar with a hemispherical electron analyzer (Specs Phoibos 150, Berlin, Germany). Core-level spectra were collected at normal emission (60° incidence) geometry and with a 0.5 eV total resolution. A gold mesh was used for the calibration of the incident photon flux. Voigt profiles were used for fitting of the C 1s and N 1s spectra. A Shirley function was used for background correction. Near-edge X-ray absorption (NEXAFS) measurements were performed in the Auger-electron yield mode near the C and N K-edges at <0.4 eV resolution. An Ar-sputtered gold foil was used for calibration and for the intensity normalization.

Raman characterization was carried out on a Jasco NRS-3100 (Tokyo, Japan) spectrophotometer with a 532 nm laser and a bandstop filter to eliminate the Rayleigh scattering. Sample points were collected every 8 cm^−1^ from 400 to 3000 cm^−1^. Spectra were analyzed with Fityk 1.3.1. The best fit was achieved with 4 Voigt functions, during which peak positions were allowed to change by <5 cm^−1^.

Elemental analysis was carried out on a Perkin Elmer 2400 (Llantrisant, UK). Samples were weighted and folded in standard tin capsules. Quartz tubes were filled with cuprin and EA-6000 and placed in the combustion furnace. The operating temperature was 975 °C. Air and helium (99.9995%) were supplied from a compressor and gas tank, respectively.

Specific surface area was determined by means of a Quantachrome Nova 1200e (Anton Paar, St Albans, UK) surface-area analyzer. Prior to measuring, the samples were degassed for 15 h at 473 K. Samples were measured with nitrogen at 77 K. To determine the specific surface area of the samples under investigation, the BET method was used. The BJH method (Barret, Joyner, and Halenda) was used to assess the pore-size distribution.

### 2.4. Electrochemical Characterization

The electrocatalytic activity of the nitrogen-doped carbon samples was evaluated under controlled hydrodynamic conditions in a three-electrode glass-cell setup. A potentiostat (Origalys model OrigaFlex-OGF01A) was used to perform the electrochemical measurements. Nitrogenated carbon samples were placed on the surface of glassy carbons from a 5 μL drop of an ultrapure water ink (3.5 mg/mL concentration). The working electrode was represented by the functionalized carbon samples deposited on the glassy-carbon rotating-disk electrode (RDE, Rillieux-la-Pape, Origalys) with a 3 mm diameter. The reference electrode was a reversible hydrogen electrode (RHE, Hydroflex^®^, Gaskatel, Kassel, Germany). To avoid Pt contamination, a graphite rod was used as the counter electrode. GC disks were cleaned by polishing with a 0.8 μm alumina slurry on a synthetic fiber cloth (NWF+, Presi, Le Cocle, Switzerland), rinsed with ultrapure water, and then polished with a 0.04 μm alumina slurry on a microcloth (DBM, Presi, Le Cocle, Switzerland). The GC disks were rinsed in ultrapure water, and then ultrasonicated in fresh ultrapure water for 5 min. The electrochemical cell was kept at 80 °C overnight in ultrapure water prior to use. All electrochemical measurements were carried out in 0.1 M KOH aqueous solutions. The working electrode was conditioned by running 10 cyclic voltammograms (CVs) between 50 mV and 1200 mV versus RHE at a 100 mV s^−1^ scan rate in N_2_ saturated 0.1 M KOH. An additional three CVs were recorded at 50 mV and 1200 mV versus RHE at 20 mV s^−1^ scan rate. Oxygen reduction reactions (ORRs) were recorded at 0, 400, 900, and 1600 rpm in O_2_-saturated 0.1 M KOH. In order to correct for the capacitive current, the CVs in nitrogen were subtracted from the ORR sweeps. The as-obtained potentials were further corrected for uncompensated solution resistance determined from electrochemical impedance spectroscopy measurements. Electrochemical degradation was carried out for 6 h in 0.1 M KOH between 0.05 and 1.20 V vs. RHE at 0.1 V s^−1^.

## 3. Results and Discussion

Textural properties were assessed by nitrogen adsorption/desorption isotherms in line with the IUPAC recommendations [26,27] (Appendix A). Investigated samples showed a surface area of 240 m^2^g^−1^, 192 m^2^g^−1^, 204 m^2^g^−1^, and 174 m^2^g^−1^ for Vulcan, CN100, CN250, and CN350, respectively. It was evident that the RF plasma treatment decreased the surface area by approximately 20% for all the samples. The BJH method was used to measure the pore diameter. Average pore-size values of 3.05 nm, 3.39 nm, 3.45 nm, and 3.39 nm were determined for Vulcan, CN100, CN250, and CN350, respectively. The smallest pore size corresponded to the sample with the largest surface area; that is, Vulcan. There was a less than 2% difference between the pore sizes of nitrogenated samples. It can be inferred that the nitrogenated carbons had similar textural properties, considering the small differences in pore size and surface area. The focus was then turned to other characteristics, such as the carbon organization.

Samples were obtained by surface treatment in a home-built cold RF plasma reactor in a nitrogen atmosphere. Elemental analysis showed an N/C ratio of 0, 2.1%, 4.7%, and 1.7% for Vulcan, CN100, CN250, and CN350, respectively. The nitrogen concentration increased linearly with the plasma power up to 250 W. At 350 W, the nitrogen concentration was smaller than at 250 W. Most probably, the average kinetic energy of the nitrogen ionized was too high, which led to scattering. Although the study of nitrogen concentration into carbon materials is interesting, it falls beyond the general scope of this study.

Synchrotron radiation photoemission spectroscopy (SRPES) was used to evaluate the nitrogen chemistry on the surface of the electrodes. The core-level peaks analyzed by synchrotron radiation had a considerably smaller full width–half maximum, which was leveraged for the multicomponent deconvolution of N 1s. The wide spectrum (Appendix A) showed that there was only carbon, nitrogen, and oxygen present on the surface. Core-level spectra of C 1s and O 1s were not fitted due to the presence of adventitious carbon and its inherent oxygen, which can easily mislead the fitting. Moreover, there were many carbon–oxygen and carbon–nitrogen overlapping moieties, which must be considered when fitting C 1s; this is still a matter of debate in the literature [15,28]. To eliminate any possible artefacts, the attention was focused only on the N 1s peak—a signal that can arise only from the sample.

Three peaks were observed in the N 1s core-level spectra: pyridinic (i.e., nitrogen in a six-membered ring), pyrrolic (i.e., nitrogen in a five-membered ring) and graphitic (i.e., nitrogen bonded to 3 carbon atoms) [29,30]. The fitting of the N 1s can be found in Appendix A. There is a large body of work that relates the ORR activity to pyridinic or graphitic, or a mix of the two [8,9,13,14,16]. The potential at 0.1 mAcm^−2^, also known as the onset potential, was plotted against the nitrogen surface concentration (Figure 1 and Appendix A). The largest onset potential was determined for CN250: approximately 0.72 V vs. RHE. The concentration of pyridinic increased from CN100 to CN350. The concentration of pyrrolic and graphitic decreased from CN100 to CN350 (Figure 1). It was expected for at least 1 nitrogen concentration ratio to have a maximum or minimum for the CN250. However, there was no clear dependence found between the nitrogen surface chemistry and onset potential.

Raman spectroscopy on amorphous nanocarbons has been well studied in the literature. It is generally accepted that there are four bands: the D band at approximately 1350 cm^−1^, the G band at approximately 1600 cm^−1^, the A band at 1500 cm^−1^, and the I band at approximately 1200 cm^−1^. A thorough discussion on the rationale of every band can be found in the three-stage model proposed by Ferrari and Robertson [31]. Peak fitting was performed with 4 Voigt functions. Further details on the fitting parameters can be found in the Appendix A. All the samples had A_D_/A_G_ ratios (i.e., area ratios) between 2.19 and 3.19, suggesting a strong amorphous character. Similar values were obtained for Vulcan [32] and nitrogen-doped carbons [30]. The average lateral dimension of the graphitic cluster (L_a_) is inversely proportional to the A_D_/A_G_ ratio (Appendix A) [33,34,35]. To put it simply, the sample with the lowest I_D_/I_G_ (i.e., 2.19 for Vulcan) possessed the highest L_a_; that is, 8.21 nm. The nitrogenated samples had a higher I_D_/I_G_ ratio than the non-nitrogenated sample (i.e., Vulcan), which implies a lower graphitization. A similar trend has been observed on other nanocarbons [30]. The first-order Raman (i.e., 1000–2000 cm^−1^) had 2 minor peaks, A and I, which were assigned to both trigonal and tetragonal carbons [36]. In this respect, the area of the A peak was normalized to the area of the D and G peaks, respectively.

Three carbon-organization characteristics (Figure 2) were plotted against the onset potential: L_a_ (that is, inversely proportional to I_D_/I_G_), A_A_/A_D_ (area ratio) and A_A_/A_G_ (area ratio). CN250 had the highest L_a_ among the nitrogen-doped samples. The lowest A_A_/A_D_ and A_A_/A_G_ ratio was determined for CN250 among the nitrogenated samples. One might be tempted to conclude that these carbon characteristics were better ORR descriptors for nanocarbons than the nitrogen surface chemistry. However, the non-nitrogenated carbon had a higher L_a_ and a lower A_A_/A_D_ and A_A_/A_G_ than CN250. If the non-nitrogenated sample is considered, then the conclusion cannot be deemed valid (Figure 2).

### 3.1. What Is the Reason behind ORR Activity in Nitrogen-Doped Carbons?

There was no clear standalone ORR descriptor for the samples tested in this work. It is clear from Figure 1 that nitrogen was mandatory to drive the ORR on carbon electrodes. The graphitization degree (Figure 2) was also a critical parameter that should be considered in designing future nanocarbon electrocatalysts. To sum up, none of the characteristics mentioned above can be viewed as a unique descriptor, albeit the combination of nitrogen surface chemistry and graphitization degree can offer guidance. Looking at the data in Figure 1 and Figure 2, it can be easily inferred that a high ORR activity included a combination of factors. To answer the question at the beginning of this subsection, one has to look beyond nitrogen surface chemistry and carbon organization.

Near-edge X-ray absorption fine structure (NEXAFS) was used to characterize the local structure of carbon and nitrogen (Appendix A). NEXAFS has been extensively used to characterize the carbon nanostructures, but this will not be detailed in the work at hand [37,38,39,40]. Similar to SRPES, peak fitting in NEXAFS is subject to speculation. To further understand the difference between the materials, a difference analysis on the NEXAFS spectra is shown in Figure 3. NEXAFS difference analysis [38] has been recently used to highlight the difference between nanocarbons and nitrogenated nanocarbons. The difference analysis (Figure 3A) showed a horizontal line before the K-edge of carbon (i.e., approximately 284 eV), which held for all the samples. There was no reason for the pre-edge to change, which validated the technique. A peak above the horizontal line indicates that there is an excess of that specific moiety characterized by that photon energy. For example, CN350 had the largest pyridinic concentration (Figure 2) which was evident from the increase in the peak at 398.5 eV for the CN350–CN100 and the decrease for the CN250–CN350 (Figure 3B). The above-mentioned examples further supported the validity of the NEXAFS difference analysis [37,38].

There were two regions of interest in the NEXAFS difference analysis of C K-edge (Figure 3A): (1) the region between 284 and 286 eV, which has been previously assigned to C 1s→π* in C = C; and (2) the region between 287 and 291 eV, which has been previously assigned to different oxygen functionalities, such as the C 1s→π* of C = O (286.5 eV) and COOH (288.3 eV), and the C 1s→σ* of C–OH (289.3 eV). Minor differences were found between CN350 and CN100. A considerable decrease was measured in the 284–286 eV region for CN250 when compared to CN100 and CN350. The decrease in this area suggested that the aromaticity was lost, which was further confirmed by an increase in the 287–291 eV region, specific for oxygen functionalities (vide supra). The reader should bear in mind that the region between 287 and 291 eV is also specific for different nitrogen functionalities [37,38]. The N/C concentration followed the trend CN250 > CN350 > CN100, which can also be inferred from Figure 3A.

C–N and C=N have remarkably similar photon-energy values as C−O and C=O in the C K-edge. To unravel the nitrogen local structure, the analysis of the N K-edge is paramount. The peak at 399.3 eV was specific for pyridine rings (i.e., N=C N 1s→π* transitions), and was shown to be the most abundant on the CN350 surface. In consequence, there was a negative peak and a positive peak at 399.5 eV for CN250–CN350 and CN250–CN100, respectively (blue and black curves in Figure 3B). The same rationale held for the CN350–CN100 curve, because the pyridinic concentration trend was CN350 > CN250 > CN100. Our attention then was turned to the pre-edge feature in the N K-edge. The peak centered at 398.5 eV (Figure 3B) was specific for pyridine conjugated to benzene units, such as the pyridine in quinoline, acridine, or phenazine. There was no difference between the CN350 and CN100 in the pre-edge area of the N-K edge (Figure 3B). The pre-edge feature was negative in the difference analysis of CN250–CN350 and CN250–CN100, which was surprising, considering the pyridinic trend determined from XPS (i.e., CN350 > CN250 > CN100). To reflect the pyridinic trend established by the XPS, the pyridinic nitrogen in CN250 must have an increased effective nuclear charge due to an electron-withdrawing group. Only oxygen has such ability among the composing elements; that is, carbon, nitrogen, and oxygen. Indeed, 4-hydroxy-2-pyridone and 2,3-pyridinedicarboxylic acid showed a 2 eV shift in the N 1s→π* in N=C with a photon energy between 401 and 402 eV [37,38]. A closer investigation of the 401–402 eV region in Figure 3B showed a peak in the 401–402 eV region for CN250–CN350 and CN250–CN100. When such a finding is corroborated with the pre-edge feature in the difference analysis of CN250–CN350 and CN250–CN100, then the pyridinic-oxygen moiety is unraveled. It is interesting to go back and look at the C K-edge in light of the pyridinic–oxygen moiety (Figure 3A). Now, the sharp peak at 288.3 eV can be explained on the basis of the C 1s→π* in COOH.

### 3.2. Tracking Degradation by Postmortem Raman Analysis

Postmortem studies; that is, the study of electrodes before and after electrochemical experiments, are paramount in elucidating the role of active sites. The reader should bear in mind that the nitrogen and carbon core-level spectra obtained from photoelectron spectroscopy were challenging to analyze due to the inherent presence of oxygen in high concentrations (arising from carbon corrosion and water adsorption). It is well-known that the aromatic rings open under ORR, which should change the carbon organization and, therefore, alter the Raman signal (Appendix A). However, the A_D_/A_G_ ratio changes were quite low (Table 1). Investigating the A_D_/A_G_ ratio values at EOT (end of test) showed remarkably similar values for nitrogenated carbons, which were spread between 2.76 and 3.05 (Table 1). Such small differences have been previously reported for Vulcan and Ketjenblack, although the authors showed a decrease in the A_D_/A_G_ ratio at EOT.

Another widely used Raman parameter is the area ratio of the A and G peak. The losses experienced by the A_A_/A_G_ were considerably larger than the A_D_/A_G_ ratio (Table 1). For example, the initial ratios of CN250 and CN100 were 0.898 and 0.999, respectively, and by EOT had increased to 1.152 and 1.044 for CN250 and CN100, respectively (Table 1). The changes in the A_D_/A_G_ and A_A_/A_G_ ratios are consistent with the ORR loss of the nitrogenated carbons (Table 1). The values for the A_D_/A_G_ ratio losses were considerably smaller than the values for the A_A_/A_G_ ratio loss. A similar trend could be inferred for the A_A_/A_D_ ratio. All in all, the postmortem Raman investigation revealed significant changes that occurred upon long-term ORR.

The CVs at EOT had similar onset potentials for all the samples, irrespective of nitrogen concentration (Appendix A). Table 1 shows that there was a considerable loss in the A_A_/A_G_ ratio loss upon extended ORR for the sample with the highest nitrogen concentration (i.e., CN250), which was also consistent with the largest loss in ORR performance. XPS was used to measure the nitrogen concentration decrease in carbon nitrides (>50% nitrogen) during photoelectrochemical water splitting. The low nitrogen concentrations that occurred during nitrogen doping (i.e., below 10%) made it hard to trace small changes in the nitrogen concentration due to inherent sample contamination with adventitious carbon (which increased the carbon concentration and decreased the N/C ratio). Table 1 shows that the A_A_/A_G_ ratio could be used to qualitatively assess the degradation in heteroatom-doped carbons. Further investigations are needed to fully understand the nature of the A peak (centered at 1500 cm^−1^) in heteroatom-doped carbons and its role in ORR.

## 4. Conclusions

In this article, we made use of a surface plasma functionalization to achieve a better understanding of the role of nitrogen in the oxygen reduction reaction on nitrogen-doped carbons. We found that if the plasma power was increased beyond 250 W, then nitrogen was not incorporated in the carbon structure. The graphitization degree decreased with increasing plasma power. Nitrogen concentration and surface chemistry (type of nitrogen) were found to be poor ORR standalone descriptors. Similarly, the degree of graphitization did not correlate with the measured ORR activity. The focus was turned to the role of oxygen, which was inherently present on the carbon surface. -edge X-ray absorption fine structure showed that if both oxygen and nitrogen were present in the same aromatic ring, then the onset potential for the ORR increased. This work shed light on the synergy between nitrogen and oxygen. Near-edge X-ray absorption fine structure should be used in the future to probe multiple heteroatoms, as it is the only available technique that can elucidate the synergy of multiple active sites.

## Figures and Tables

**Figure 1 nanomaterials-11-01198-f001:**
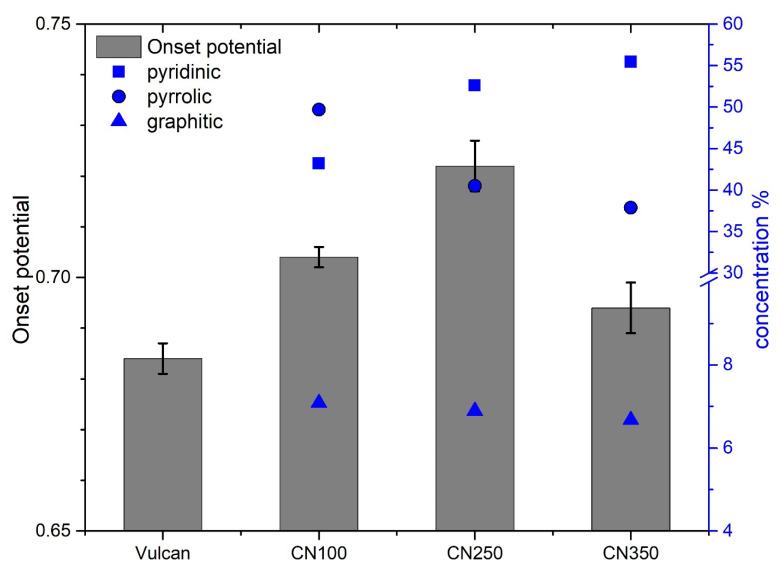
Grey column bar: onset potential (i.e., potential at 0.1 mAcm^−2^); blue squares: pyridinic surface concentration determined from the N 1s core level spectra; blue circles: pyrrolic surface concentration determined from the N 1s core level spectra; blue triangles: graphitic surface concentration determined from the N 1s core level spectra. The blue y-axis has a breaking from 10 to 30% to accommodate a better view of the graphitic concentration (blue triangles).

**Figure 2 nanomaterials-11-01198-f002:**
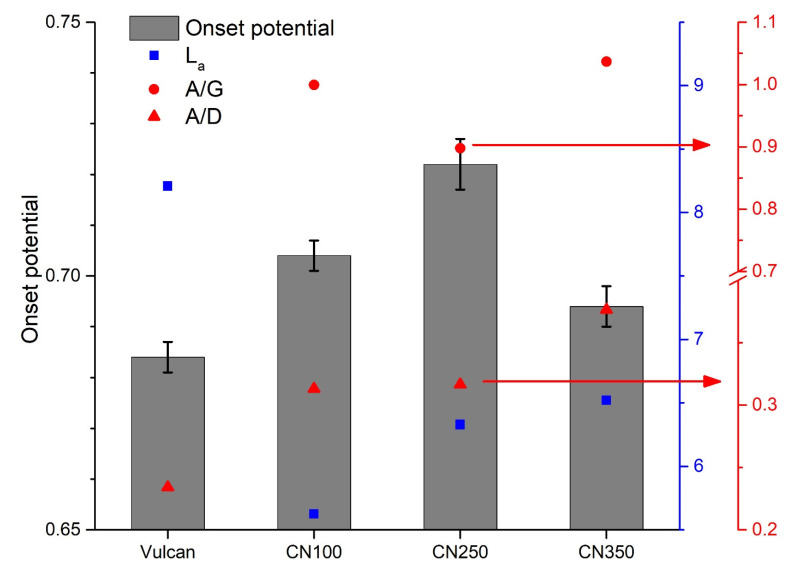
Grey column bar: onset potential (i.e., potential at 0.1 mAcm^−2^); blue squares: lateral dimension of the graphitic cluster, L_a_ (nm), which is inversely proportional to the I_D_/I_G_ ratio; red circles: area of the A band divided by the D band; red triangles: area of the A band divided by the G band.

**Figure 3 nanomaterials-11-01198-f003:**
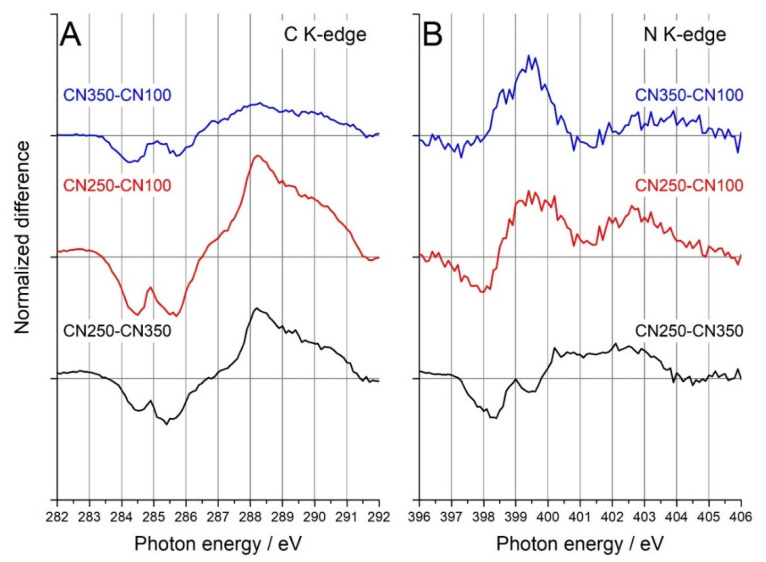
NEXAFS differential analysis at the C K-edge (**A**) and N K-edge (**B**).

**Table 1 nanomaterials-11-01198-t001:** Postmortem Raman investigation results. BOT = beginning of test; EOT = end of test. The electrochemical stability was performed for 6 h (see Experimental section). If the relative loss has a negative sign, then it can be considered as a gain.

	A_D_/A_G_			A_A_/A_G_			ORR
	BOT	EOT	Relative Loss	BOT	EOT	Relative Loss	E_1/2_ Difference/mV
**Vulcan**	2.215	2.300	−1.2%	0.512	0.528	−3.1%	−12
**CN100**	3.193	3.050	+4.5%	0.999	1.044	−4.5%	−18
**CN250**	2.838	2.896	−2.0%	0.898	1.152	−28.3%	−40
**CN350**	2.755	2.764	−0.4%	1.037	1.093	−5.5%	−25

## Data Availability

The data presented in this study are available on request from the corresponding author.

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
