# Peer review of "Beyond Nitrogen in the Oxygen Reduction Reaction on Nitrogen-Doped Carbons: A NEXAFS Investigation"

_nanomaterials, 2021, doi:10.3390/nano11051198_

Round 1

Reviewer 1 Report

In this manuscript, the authors investigated the role of nitrogen and oxygen functional groups toward oxygen reduction reaction using near-edge X-ray absorption fine structure differential analysis, and found that the ORR activity increases only when both oxygen and nitrogen are present in the same aromatic ring. I would recommend its publication if the following questions are addressed:

  1. In figure 1, the authors tried to correlate the ORR activity with the ratios of different nitrogen species detected by XPS. The ratios only represent the relative amount of different nitrogen species; however, the ORR activity is more relevant to the absolute number of active sites. Thus, it is more reasonable to compare the ORR activity with the absolute peak area.
  2. On page 5, line 201, the authors claim that "The average lateral dimension of the graphitic cluster (La) is inversely proportional to the AD/AG ratio". Could the authors explain this in detail? Generally, AD/AG is indicative of the degree of disorder of carbon. It is better to provide some references here.
  3. The authors also tried to study the structural change before and after ORR activity measurements by Raman to track the degradation mechanism. However, no clear conclusion is drawn. This is expected since the Raman spectroscopy was not able to detect the key descriptor for ORR. Instead, NEXAFS differential analysis was able to identify both oxygen and nitrogen in the same aromatic ring are required to achieve a high ORR activity. Thus, it is more reasonable to use NEXAFS differential analysis to investigate the degradation mechanism.
  4. Minor questions: please check the figure numbers in supporting information (e.g., figure S3 and S4). Also, on page 6, line 229, why there is figure S5 there?

Author Response

In this manuscript, the authors investigated the role of nitrogen and oxygen functional groups toward oxygen reduction reaction using near-edge X-ray absorption fine structure differential analysis, and found that the ORR activity increases only when both oxygen and nitrogen are present in the same aromatic ring. I would recommend its publication if the following questions are addressed:

  1. In figure 1, the authors tried to correlate the ORR activity with the ratios of different nitrogen species detected by XPS. The ratios only represent the relative amount of different nitrogen species; however, the ORR activity is more relevant to the absolute number of active sites. Thus, it is more reasonable to compare the ORR activity with the absolute peak area.

Response 1. We thank the reviewer for the suggestion, and we fully agree with it. We have replaced the Figure 1 with the figure suggested by the reviewer. We replaced the paragraph:

The concentration ratio of pyridinic/pyrrolic and graphitic/pyrrolic was increasing from CN100 to CN350. The concentration ratio of graphitic/pyridinic was decreasing from CN100 to CN350.”

with

The concentration of pyridinic was increasing from CN100 to CN350. The concentration of pyrrolic and graphitic was decreasing from CN100 to CN350.”

  1. On page 5, line 201, the authors claim that "The average lateral dimension of the graphitic cluster (La) is inversely proportional to the AD/AG ratio". Could the authors explain this in detail? Generally, AD/AG is indicative of the degree of disorder of carbon. It is better to provide some references here.

Response 2. We acknowledge that there is a missing reference here. We added 3 references at the end of the sentence. Tuinstra and Koenig showed for the first time that AD/AG = 4.4/La which has been used extensively in many carbon-based studies. A more thorough discussion on the expression can be found in the work of Puech et al. We believe that references 33-35 will guide the readers to the relation between average lateral dimension and AD/AG.

  1. The authors also tried to study the structural change before and after ORR activity measurements by Raman to track the degradation mechanism. However, no clear conclusion is drawn. This is expected since the Raman spectroscopy was not able to detect the key descriptor for ORR. Instead, NEXAFS differential analysis was able to identify both oxygen and nitrogen in the same aromatic ring are required to achieve a high ORR activity. Thus, it is more reasonable to use NEXAFS differential analysis to investigate the degradation mechanism.

Response 3. We thank the reviewer for this suggestion. The NEXAFS study was conducted at the Elettra synchrotron while the electrochemistry was conducted in our lab in Romania. The reviewer’s idea is good, and it will be considered in the future. However, the task will need to apply for beamtime, get the beamtime approved and conducting the experiment. The task will take a couple of months. Therefore, we kindly rebut this comment.

  1. Minor questions: please check the figure numbers in supporting information (e.g., figure S3 and S4). Also, on page 6, line 229, why there is figure S5 there?

Response 4. We accept the minor mistake and have corrected accordingly. We updated the supporting information and we erased fig S5 from the main manuscript.

Reviewer 2 Report

The authors describe the use of a surface plasma functionalization to understand the role of nitrogen in the oxygen reduction reaction at nitrogen doped carbons. The characterization was supported by raman spectroscopy, X-ray absorption, BET and electrochemical analysis. The manuscript is well conceived and the topic is deepened, for this reason it is suitable for publication . My only suggestion is to add a couple of references in the introduction (page 1, lines 39 and 42) and the schemes of the supporting information within the text.

Author Response

The authors describe the use of a surface plasma functionalization to understand the role of nitrogen in the oxygen reduction reaction at nitrogen doped carbons. The characterization was supported by raman spectroscopy, X-ray absorption, BET and electrochemical analysis. The manuscript is well conceived and the topic is deepened, for this reason it is suitable for publication . My only suggestion is to add a couple of references in the introduction (page 1, lines 39 and 42) and the schemes of the supporting information within the text.

Response 1. We have adapted the text at lines 39-42 accordingly: “Oxygen reduction reaction (ORR) is the key reaction in air–cathode electrochemical devices: polymer electrolyte membrane fuel cells1, metal–air batteries2, and microbial fuel cells3.”

Reviewer 3 Report

General comments Manuscript is well written and can be published as it is. But I have two suggestions. Specific comments Introduction is short and is primarily focused on N doping. Authors must also mention the beneficial effect of other heteroatoms on ORR activity. Authors must consider adding valence band analysis as outlined in the paper "Jiao, Y. et al. Origin of the electrocatalytic oxygen reduction activity of graphene-based catalysts: a roadmap to achieve the best performance. J. Am. Chem. Soc. 136, 4394–4403 (2014)"

Author Response

General comments Manuscript is well written and can be published as it is. But I have two suggestions. Specific comments Introduction is short and is primarily focused on N doping. Authors must also mention the beneficial effect of other heteroatoms on ORR activity. Authors must consider adding valence band analysis as outlined in the paper "Jiao, Y. et al. Origin of the electrocatalytic oxygen reduction activity of graphene-based catalysts: a roadmap to achieve the best performance. J. Am. Chem. Soc. 136, 4394–4403 (2014)"

Introduction is short and is primarily focused on N doping. Authors must also mention the beneficial effect of other heteroatoms on ORR activity. 

Response 1. We adapted the text as per reviewer suggestion (including the suggested reference) : "Recent theoretical studies suggested that oxygen does not bind at nitrogen atoms22 and a similar mechanism is expected for all heteroatom doped carbons. The role of other heteroatoms (other than nitrogen) in oxygen reduction reaction activity has been studied23,24. Nanocarbons are inherently doped with oxygen. Therefore, traditional heteroatom doped carbons are dual doped. A secondary atom has been shown to break the well-known scaling relationships25. There are no studies to date that characterize simultaneously the inherent heteroatom (that is oxygen) and the secondary heteroatom (that is nitrogen, phosphorous, boron, etc). Such studies are difficult as standard characterization tools focus on one heteroatom at a time."

Authors must consider adding valence band analysis as outlined in the paper "Jiao, Y. et al. Origin of the electrocatalytic oxygen reduction activity of graphene-based catalysts: a roadmap to achieve the best performance. J. Am. Chem. Soc. 136, 4394–4403 (2014)"

Response 2. We did conduct the valence band analysis and we concluded that the valence band spectra is challenging to characterize for our nanocarbons. Valence band spectra is highly surface sensitive. The surface of our nanocarbons will be quickly covered by adsorbed oxygen. Then there is the problem of morphology and the way that the spheres are stacking. To remove any questionable results and interpretations, we decided to remove the valence band data from the manuscript. We kindly rebut this comment.